# UNSUPERVISED CIPHER CRACKING USING DISCRETE GANS

**Aidan N. Gomez** [1,2]
aidan@cs.toronto.edu

**Sicong Huang** [1,2]
huang@cs.toronto.edu

**Ivan Zhang** [2]
ivan@ivanzhang.ca

**Bryan M. Li** [1,2]
bryan@bryanli.io

**Muhammad Osama** [1,2]
muhammad.osama@mcode.ca

**Łukasz Kaiser** [3]
lukaszkaiser@google.com

[1] Department of Computer Science, University of Toronto
[2] FOR.ai
[3] Google Brain

## ABSTRACT

This work details CipherGAN, an architecture inspired by CycleGAN used for inferring the underlying cipher mapping given banks of unpaired ciphertext and plaintext. We demonstrate that CipherGAN is capable of cracking language data enciphered using shift and Vigenère ciphers to a high degree of fidelity and for vocabularies much larger than previously achieved. We present how CycleGAN can be made compatible with discrete data and train in a stable way. We then prove that the technique used in CipherGAN avoids the common problem of uninformative discrimination associated with GANs applied to discrete data.

## 1 INTRODUCTION

Humans have been encoding messages for secrecy since before the ancient Greeks, and for the same amount of time, have been fascinated with trying to crack these codes using brute-force, frequency analysis, crib-dragging and even espionage. Simple ciphers have, in the past century, been rendered irrelevant in favor of the more secure encryption schemes enabled by modern computational resources. However, the question of cipher-cracking remains an interesting problem since it requires an intimate understanding of the structure in a language. Nearly all automated cipher cracking techniques have had to rely on a human-in-the-loop; grounding the automated techniques in a human's preexisting knowledge of language to clean up the errors made by simple algorithms such as frequency analysis. Across a number of domains, the use of hand-crafted features has often been replaced by automatic feature extraction directly from data using end-to-end learning frameworks (Goodfellow et al., 2016). The question to be addressed is as follows:

*Can a neural network be trained to deduce withheld ciphers from unaligned text, without the supplementation of preexisting human knowledge?*

The implications for such a general framework would be far-reaching in the field of unsupervised translation, where each language can be treated as an enciphering of the other. The decoding of the Copiale cipher (Knight et al., 2011) stands as an excellent example of the potential for machine learning techniques to decode enciphered texts by treating the problem as language translation. The CycleGAN (Zhu et al., 2017) architecture is extremely general and we demonstrate our adaptation, CipherGAN, is capable of cracking ciphers to an extremely high degree of accuracy. CipherGAN

---

Code available at: github.com/for-ai/ciphergan

requires little or no modification to be applied to plaintext and ciphertext banks generated by the user's cipher of choice.

In addition to presenting a GAN that can crack ciphers, we contribute the following techniques:

- We show how to stabilize CycleGAN training: our CipherGAN achieves good performance in all training runs, compared to approximately 50% of runs for the original CycleGAN.

- We provide a theoretical description and analysis of the uninformative discrimination problem that impacts GANs applied to discrete data.

- We introduce a solution to the above problem by operating in the embedding space and show that it works in practice.

## 1.1 SHIFT AND VIGENÈRE CIPHERS

The shift and Vigenère ciphers are well known historical substitution ciphers. The earliest known record of a substitution cipher is believed to have dated back to 58 BCE, when Julius Caesar replaced each letter in a message with the letter that was three places further down the alphabet (Singh, 2000).

| Plain alphabet | a b c d e f g h i j k l m n o p q r s t u v w x y z |
|---|---|
| Shifted alphabet | D E F G H I J K L M N O P Q R S T U V W X Y Z A B C |

Figure 1: An example of a right-shift-3 cipher.

Using Figure 1, the message "*attackatdawn*" can be encrypted to "*DWWDFNDWGDZQ*". This message can be easily deciphered by the intended recipient (who is aware of the particular shift number used) but looks meaningless to a third party. The shift cipher ensured secure communication between sender and receiver for centuries, until the ninth century polymath Al-Kindi introduced the concept of frequency analysis (Singh, 2000). He suggested that it would be possible to crack a cipher simply by analyzing the individual characters' frequencies. For instance, in English the most frequently occurring letters are '*e*' (12.7%), '*t*' (9.1%) and '*a*' (8.2%); whereas '*q*', '*x*' and '*z*' each have frequency of less than 1%. Moreover, the code-breaker can also focus on bigrams of repeated letters; '*ss*', '*ee*', and '*oo*' are the most common in English. This structure in language provides an exploit of efficiency to the code-breaker.

Polyalphabetic substitution ciphers, including the Vigenère cipher, were introduced to inhibit the use of n-gram frequency analysis in determining the cipher mapping. Instead, the encrypter further scrambles the message by using a separate shift cipher for each element of a key that is tiled to match the length of the plaintext. Increasing the key length greatly increases the number of possible combinations and thus prevents against basic frequency analysis. In the mid nineteenth century, Charles Babbage recognized that the length of the used key could be determined by counting the repetitions and spacing of sequences of letters in the cipher (Singh, 2000). Using the determined length, we can then apply frequency analysis on the index of the cipher base. This method makes it possible to break the Vigenère cipher, but is very time consuming and requires strong knowledge of the language itself.

There is a rich literature of automated shift-cipher cracking techniques (Ramesh et al., 1993; Forsyth & Safavi-Naini, 1993; Hasinoff, 2003; Knight et al., 2006; Verma et al., 2007; Raju et al., 2010; Knight et al., 2011) many of which achieve excellent results which is what one would expect from hand-crafted algorithms targeting specific ciphers and vocabularies. Work on automated cracking of polyalphabetic ciphers (Carroll & Martin, 1986; Toemeh & Arumugam, 2008; Omran et al., 2011) has seen similar success on small vocabularies. It is a difficult matter to compare the results of previous work with our own as their focus ranges from inferring cipher keys (Carroll & Martin, 1986; Ramesh et al., 1993; Omran et al., 2011), to inferring the mappings given limited quantities of ciphertext (determining unicity distance) (Carroll & Martin, 1986; Ramesh et al., 1993; Hasinoff, 2003; Verma et al., 2007), to analyzing the unicity distance required to solve small percentages of the cipher mappings (i.e. 20% in Carroll & Martin (1986)).

In comparison to these past works, we afford ourselves the advantage of an unconstrained corpus of ciphertext, however, we prescribe ourselves the following constraints: our model is not provided any prior knowledge of vocabulary element frequencies; and, no information about the cipher key is

provided. Another complexity our work must overcome is our significantly larger vocabulary sizes; all previous work has addressed vocabularies of approximately 26 characters, while our model is capable of solving word-level ciphers with over 200 distinct vocabulary elements. As such, our methodology is notably 'hands-off' in comparison to previous work and can be easily applied to different forms of cipher, different underlying data and unsupervised text alignment tasks.

## 1.2 GANs and Wasserstein GANs

Generative Adversarial Networks (GANs) are a class of neural network architectures introduced by Goodfellow et al. (2014) as an alternative to optimizing likelihood under a true data distribution. Instead, GANs balance the optimization of a generator network which attempts to produce convincing samples from the data distribution, and a discriminator which is trained to distinguish between samples from the true data distribution and the generator's synthetic samples. GANs have been shown to produce compelling results in the domain of image generation, but comparatively weak performance in domains using discrete data (discussed in Section 2).

The original GAN discriminator objective as introduced in Goodfellow et al. (2014) is:

$$D^* = \arg\max_D \mathbb{E}_{x \sim \mathcal{X}}[\log D(x)] - \mathbb{E}_{z \sim \mathcal{Z}}[\log(1 - D(F(z)))] \tag{1}$$

Where $F$ is the generator network and $D$ is the discriminator network. This loss is vulnerable to the problem of 'mode collapse' where the generative distribution collapses to produce a generating distribution with low diversity. In order to more broadly distribute the mass, the Wasserstein GAN (WGAN) objective (Arjovsky et al., 2017) considers the set of K-Lipschitz discriminator functions $D : X \to \mathbb{R}$ and minimizes the earth movers (1st Wasserstein) distance. The Lipschitz condition is enforced by clipping discriminators weights to fall within a predefined range.

$$D^* = \arg\max_{\|D\|_L \leq K} \mathbb{E}_{x \sim \mathcal{X}}[D(x)] - \mathbb{E}_{z \sim \mathcal{Z}}[D(F(z))] \tag{2}$$

An improved WGAN objective, introduced by Gulrajani et al. (2017), enforced the Lipschitz condition using a Jacobian regularization term instead of the originally proposed weight-clipping solution. This resulted in more stable training, avoiding capacity under-use and exploding gradients, and improved network performance over weight-clipping.

$$D^* = \arg\max_D \mathbb{E}_{x \sim \mathcal{X}}[D(x)] - \mathbb{E}_{z \sim \mathcal{Z}}[D(F(z))] + $$
$$\alpha \cdot \mathbb{E}_{\hat{x} \sim \hat{\mathcal{X}}}[(\|\nabla_{\hat{x}} D(\hat{x})\|_2 - 1)^2] \tag{3}$$

Here $\hat{\mathcal{X}}$ are samples taken along a line between the true data distribution $\mathcal{X}$ and the generator's data distribution $\mathcal{X}_g = \{F(z) | z \sim \mathcal{Z}\}$.

## 1.3 CycleGAN

CycleGAN (Zhu et al., 2017) is a generative adversarial network designed to learn a mapping between two data distributions without supervision. Three separate works (Zhu et al., 2017; Yi et al., 2017; Liu et al., 2017) share many of the core features we describe below, however, for simplicity we will refer to CycleGAN as the basis for our work as it is the most similar to our model. It acts on distributions $\mathcal{X}$ and $\mathcal{Y}$ by using two mapping generators: $F : \mathcal{X} \to \mathcal{Y}$ and $G : \mathcal{Y} \to \mathcal{X}$; and two discriminators: $D_{\mathcal{X}} : \mathcal{X} \to [0, 1]$ and $D_{\mathcal{Y}} : \mathcal{Y} \to [0, 1]$.

CycleGAN optimizes the standard GAN loss $\mathcal{L}_{\text{GAN}}$:

$$\mathcal{L}_{\text{GAN}}(F, D_{\mathcal{Y}}, \mathcal{X}, \mathcal{Y}) = \mathbb{E}_{y \sim \mathcal{Y}}[\log D_{\mathcal{Y}}(y)] + \mathbb{E}_{x \sim \mathcal{X}}[\log(1 - D_{\mathcal{Y}}(F(x)))] \tag{4}$$

While also considering a reconstruction loss, or 'cycle' loss $\mathcal{L}_{\text{cyc}}$:

$$\mathcal{L}_{\text{cyc}}(F, G, \mathcal{X}, \mathcal{Y}) = \mathbb{E}_{x \sim \mathcal{X}}[\|G(F(x)) - x\|_1] + \mathbb{E}_{y \sim \mathcal{Y}}[\|F(G(y)) - y\|_1] \tag{5}$$

Taken together the losses are balanced using a hyperparameter $\lambda$:

$$\mathcal{L}(F, G, D_{\mathcal{X}}, D_{\mathcal{Y}}, \mathcal{X}, \mathcal{Y}) = \mathcal{L}_{\text{GAN}}(F, D_{\mathcal{Y}}, \mathcal{X}, \mathcal{Y}) + \mathcal{L}_{\text{GAN}}(G, D_{\mathcal{X}}, \mathcal{Y}, \mathcal{X}) + \lambda \cdot \mathcal{L}_{\text{cyc}}(F, G, \mathcal{X}, \mathcal{Y})$$

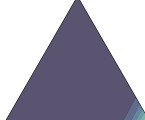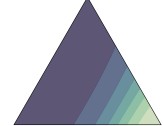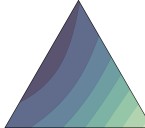

Figure 2: Discriminators trained on the toy example of recognizing the bottom-right corner of a simplex as true data. From left to right the discriminators were regularized using: nothing; WGAN Jacobian norm regularization; and, the relaxed sampling technique.

This leads to the training objectives:

$$F^* = \arg\min_F \mathcal{L}_{\mathrm{cyc}}(F, G, \mathcal{X}, \mathcal{Y}) + \mathcal{L}_{\mathrm{GAN}}(F, D_{\mathcal{Y}}, \mathcal{X}, \mathcal{Y})$$

$$G^* = \arg\min_G \mathcal{L}_{\mathrm{cyc}}(F, G, \mathcal{X}, \mathcal{Y}) + \mathcal{L}_{\mathrm{GAN}}(G, D_{\mathcal{X}}, \mathcal{Y}, \mathcal{X})$$

$$D_{\mathcal{X}}^* = \arg\max_{D_{\mathcal{X}}} \mathcal{L}_{\mathrm{GAN}}(G, D_{\mathcal{X}}, \mathcal{Y}, \mathcal{X})$$

$$D_{\mathcal{Y}}^* = \arg\max_{D_{\mathcal{Y}}} \mathcal{L}_{\mathrm{GAN}}(F, D_{\mathcal{Y}}, \mathcal{X}, \mathcal{Y})$$

CycleGAN uses $\mathcal{L}_{\mathrm{cyc}}$ to avoid mode collapse by preserving reconstruction of mapping inputs from outputs. It has demonstrated excellent results in unpaired image translation between two visually similar categories. Our architecture is the first example of this unsupervised learning framework being successfully applied to discrete data such as language.

## 2 DISCRETE GANS

Applying GANs to discrete data generation is still an open research problem that has seen great interest and development. The primary difficulty with training discrete data generators in a GAN setting is the lack of a gradient through a discrete node in the computation graph. The alternatives to producing discrete outputs - for instance, generators producing a categorical distribution over discrete elements - are prone to *uninformative discrimination* (described below), in that, the discriminator may use an optimal discrimination criterion that is unrelated to the correctness of the re-discretized generated data. In our example of a continuous distribution over discrete elements, the produced samples all lie within the standard simplex $\Delta^k$ with dimension $k$ equal to the number of elements in the distribution. In this case, samples from the true data distribution always lie on a vertex $\mathbf{v}_i$ of the simplex, while any sub-optimal generator will produce samples within the simplex's interior $\Delta^k \backslash \{\mathbf{v}_1, ..., \mathbf{v}_k\}$. In this example, a discriminator which performs uninformative discrimination might evaluate a sample's membership in the vertices of the simplex as an optimal discrimination criterion, which is entirely uninformative of the correctness of re-discretized samples from the generator.

A number of solutions to training generators with discrete outputs have been proposed: SeqGAN (Yu et al., 2017) uses the REINFORCE gradient estimate to train the generator; Boundary-seeking GANs (Hjelm et al., 2017) and maximum-likelihood augmented GANs (Che et al., 2017) proposed a gradient approximation with low bias and variance that resembles the REINFORCE (Williams, 1992) estimator. Gumbel-softmax GANs (Kusner & Hernández-Lobato, 2016) replace discrete variables in the simplex with continuous relaxations called Concrete (Maddison et al., 2016) or Gumbel-softmax (Jang et al., 2016) variables; WGANs (Arjovsky et al., 2017) were suggested as a remedy to the uninformative discrimination problem by ensuring that the discriminator's rate of change with respect to its input is bound by some constant.

Our work utilizes both the Wasserstein GAN (Gulrajani et al., 2017; Arjovsky et al., 2017) and relaxation of discrete random variables (such as Concrete/Gumbel-softmax). It has been noted multiple times in implementations of CycleGAN as well as in the original paper itself that the architecture was sensitive to initialization and requires repeat attempts in order to converge to a satisfactory mapping (Bansal & Rathore, 2017; Sari, 2017). Our architecture suffered the same instability before the WGAN Jacobian norm regularization term was added to the discriminator's loss. In addition, we found that having the discriminator operate over embedding space instead of directly over softmax vectors produced by our generator has improved performance.

Our hypothesis, which is justified by Proposition 1 below, is that the embedding vectors may act as continuous relaxations of discrete random variables as small, noisy updates are applied throughout training; Proposition 1 asserts that by replacing discrete random variables with continuous ones, our discriminator is prevented from arbitrarily approximating a Dirac delta distribution. Figure 2 shows simple discriminators trained on the toy task of identify a single vertex of a simplex as true data; it is clear that a lack of regularization leads to the discriminator collapsing to the vertex of the simplex, leaving approximately zero gradient everywhere; while the Jacobian regularization of Wasserstein GANs leads to the space covered by lines leading from the true data vertex to the generated data having a lower rate of change (note that the gradient is still close to zero in the remaining area of the simplex); and finally, replacing the discrete random variables of the true data with continuous samples about the vertex results in a much more gradual transition, which is desirable since it provides a stronger gradient signal from which to learn.

Unique to CycleGAN is the auxiliary cycle loss described in Section 1.3. The effect of this additional objective is the generated samples regularly being forced away from the discriminator's minimum in favor of a mapping that better-satisfies reconstruction. For instance, it may be the case that the discriminator favors a particular cipher mapping that is not bijective; in this case, the model will receive a strong signal from the cycle loss *away* from the discriminator's minimum. In these cases where the model moves against the gradient it receives from the discriminator it may be the case that this region has near zero curvature (as is visually discernible from Figure 2); this is because the WGAN curvature regularization (see Equation 3) has not been applied in this region.

'Curvature' here refers to the curvature of the discriminator's output with respect to its input; this curvature determines the strength of the training signal received by the generator. Low curvature means little information for the generator to improve itself with. This motivates the benefits of having strong curvature globally, as opposed to linearly between the generators samples and the true data. Kodali et al. (2017) proposes regularizing in all directions about the generated samples, which would likely remedy the vanishing gradient in our case as well; for our experiments, the relaxed sampling technique proved effective. It should also be noted that Luc et al. (2016) propose something similar to relaxed sampling whereby they replace the ground-truth discrete tokens with a distribution over the vocabulary that distributes some of the mass across the remaining incorrect tokens.

Let us now introduce the definitions needed for the formal presentation of Proposition 1.

**Definitions.**

- (Continuous relaxation of a discrete set). *A continuous relaxation of a discrete set $\mathcal{X}$ is a proper, path-connected metric space $\overline{\mathcal{X}}$ satisfying $\mathcal{X} \subset \overline{\mathcal{X}}$.*

- (Rediscretization function). *A rediscretization function is an injective function $R : \overline{\mathcal{X}} \to \mathcal{X}$ from a continuous relaxation $\overline{\mathcal{X}}$ of discrete space $\mathcal{X}$ satisfying $\forall x \in \mathcal{X}, \exists \epsilon > 0$ s.t. $R \equiv x$ on $B_\epsilon[x]$. Note that $R$ defines an equivalence relation in $\overline{\mathcal{X}}$.*

- (Uninformative Discrimination). *A discriminator $D_\mathcal{X}$ is said to perform uninformative discrimination under rediscretization function $R$ if: $\exists x \in \mathcal{X}, \bar{x} \in \overline{\mathcal{X}}$ s.t. $(R(\bar{x}) = x) \wedge (D_\mathcal{X}(x) \not\approx D_\mathcal{X}(\bar{x}))$.*

- (Continuous relaxation of a function). *A continuous relaxation of a function over discrete sets $F : \mathcal{X} \to \mathcal{Y}$ is another function $\overline{F} : \overline{\mathcal{X}} \to \overline{\mathcal{Y}}$ (where $\overline{\mathcal{X}}, \overline{\mathcal{Y}}$ are continuous relaxations of $\mathcal{X}, \mathcal{Y}$) such that $\overline{F}$ is continuous and $\overline{F}(x) = F(x), \forall x \in \mathcal{X}$.*

The following proposition (proved in the Appendix) forms the theoretical basis of the technique.

**Proposition 1** (Reliable Fooling Via Relaxation)**.**
*Given:*

- *discrete spaces $\mathcal{X}, \mathcal{Y}$ and continuous relaxations $\overline{\mathcal{X}}, \overline{\mathcal{Y}}$*

- *generators $F : \overline{\mathcal{X}} \to \overline{\mathcal{Y}}, G : \overline{\mathcal{Y}} \to \overline{\mathcal{X}}$ bijections satisfying $F = G^{-1}$*

- *discrete discriminators $D_\mathcal{X}, D_\mathcal{Y}$ both optimal for fixed $F, G$*

- *rediscretization functions $R_\mathcal{X}, R_\mathcal{Y}$*

*Suppose: F is approximately volume preserving in a small region about each $x \in \mathcal{X}$. Consequently the same is true for G about each $y \in \mathcal{Y}$.*
*If: during training, we replace discrete random variables from $\mathcal{X}$ which lie in the continuous metric space $\overline{\mathcal{X}}$ with samples from regions about them.*
*Then: the optimal relaxed discriminators $D_{\overline{\mathcal{X}}}$ and $D_{\overline{\mathcal{Y}}}$ have a non-empty region about each $x \in \mathcal{X}$ and $y \in \mathcal{Y}$ where they are expected to assign values close to $D_{\mathcal{X}}(x)$ and $D_{\mathcal{Y}}(y)$.*

Figure 3 compares a model trained with embedding vectors versus one with only the softmax outputs. It becomes clear on a harder task, such as Vigenère, that the embeddings vastly outperforms softmax in terms of speed of convergence and final accuracy; however we found that the simpler task of a shift cipher showed little difference between embeddings and softmax, suggesting an increase in task complexity increases the benefits provided by the stronger gradient signal of embeddings.

# 3 METHOD

## 3.1 CIPHERGAN

GANs applied to text data have yet to produce truly convincing results (Kawthekar et al.). Previous attempts at discrete sequence generation with GANs have generally utilized a generator outputting a probability distribution over the token space (Gulrajani et al., 2017; Yu et al., 2017; Hjelm et al., 2017). This leads to the discriminator receiving a sequence of discrete random variables from the data distribution, and a sequence of continuous random variables from the generator distribution; making the task of discrimination trivial and uninformative of the underlying data distribution. In order to avoid such a scenario, we perform all discrimination within the embedding space by allowing the generator's output distribution to define a convex combination of corresponding embeddings. This leads to the following losses:

$$\mathcal{L}_{\text{GAN}}(F, D_{\mathcal{Y}}, \mathcal{X}, \mathcal{Y}) = \mathbb{E}_{y \sim \mathcal{Y}}[\log D_{\mathcal{Y}}(y \cdot W_{Emb}^{\top})]$$
$$+ \mathbb{E}_{x \sim \mathcal{X}}[\log(1 - D_{\mathcal{Y}}(F(x \cdot W_{Emb}^{\top}) \cdot W_{Emb}^{\top}))]$$
$$\mathcal{L}_{\text{cyc}}(F, G, \mathcal{X}, \mathcal{Y}) = \mathbb{E}_{x \sim \mathcal{X}}[\|G(F(x \cdot W_{Emb}^{\top}) \cdot W_{Emb}^{\top}) - x\|_1]$$
$$+ \mathbb{E}_{y \sim \mathcal{Y}}[\|F(G(y \cdot W_{Emb}^{\top}) \cdot W_{Emb}^{\top}) - y\|_1]$$

We perform an inner product between the embeddings $W_{Emb}$ and the one-hot vectors in $x$ as well as between the embeddings and the softmax vectors produced by generators $F$ and $G$. The former is equivalent to a lookup operation over the table of embedding vectors, while the latter is a convex combination between all vectors in the vocabulary. The embeddings $W_{Emb}$ are trained at each step to minimize $\mathcal{L}_{\text{cyc}}$ and maximize $\mathcal{L}_{\text{GAN}}$, meaning the embeddings are easily mapped from and are easy to discriminate. As was discussed in Section 2, training with the above loss functions was unstable, with approximately three of every four experiments failing to produce compelling results. This is a problem we observed with the original CycleGAN horse-zebra experiment, and one that has been noted by multiple re-implementations online (Bansal & Rathore, 2017; Sari, 2017). We were able to significantly increase the stability by training the discriminator loss along with the Lipschitz conditioning term from the improved Wasserstein GAN (Gulrajani et al., 2017) (see Equation 3 and Fedus et al. (2017)), resulting in the following loss (DualGAN (Yi et al., 2017) opted to use weight-clipping to enforce the Lipschitz condition):

$$\mathcal{L}_{\text{GAN}}(F, D_{\mathcal{Y}}, \mathcal{X}, \mathcal{Y}) = \mathbb{E}_{y \sim \mathcal{Y}}[D_{\mathcal{Y}}(y \cdot W_{Emb}^{\top})]$$
$$- \mathbb{E}_{x \sim \mathcal{X}}[D_{\mathcal{Y}}(F(x \cdot W_{Emb}^{\top}) \cdot W_{Emb}^{\top})]$$
$$+ \alpha \cdot \mathbb{E}_{\hat{y} \sim \hat{y}}[(\|\nabla_{\hat{y}} D_{\mathcal{Y}}(\hat{y})\|_2 - 1)^2]$$

As a consequence of Proposition 1, discriminators trained on non-stationary embeddings will be unable to approximate Dirac delta distributions to arbitrary accuracy; implying there are dedicated 'safe-zones' about members of $\mathcal{X}$ where the generator can reliably fool the discriminator and uninformative discrimination is prevented.

In our experiments, we jointly train the embedding vectors as parameters of the model. The gradient updates applied to these vectors introduces noise between training iterations; we observed that embedding vectors tend to remain in a bounded region after the initial steps of training. We found

| Work | Ciphertext Length | Accuracy |
|------|-------------------|----------|
| Hasinoff (2003) | 500 | $\sim 97\%$ |
| Forsyth & Safavi-Naini (1993) | 5000 | $\sim 100\%$ |
| Ramesh et al. (1993) | 160 | $\sim 78.5\%$ |
| Verma et al. (2007) | 1000 | $\sim 87\%$ |

Table 1: Previous results on automated shift cipher cracking with limited ciphertext length.

that simply replacing the data with embedding vectors had a similar effect to performing the random sampling described in Proposition 1 (see Figure 3).

## 4 EXPERIMENTS

### 4.1 DATA

Our experiments use plaintext natural language samples from the Brown English text dataset (Francis & Kucera, 1979). We generate $2 * \texttt{batch\_size}$ plaintext samples, the first half are fed as the CycleGAN's $\mathcal{X}$ distribution and the second half is passed through the cipher of choice and fed as the $\mathcal{Y}$ distribution.

For our natural language plaintext data we used the Brown English-language corpus which consists of over one million words in 57340 sentences. We experiment with both word-level "Brown-W" and character-level "Brown-C" vocabularies. For word-level vocabularies, we control the size of the vocabulary by taking the top $k$ most frequent words and introducing an 'unknown' token which we use to replace all words that are not within the taken vocabulary. We demonstrate our method's ability to scale to large vocabularies using the word-level vocabularies; more modern enciphering techniques rely on large substitution-boxes (S-boxes) with many (often hundreds of) elements.

### 4.2 TRAINING

As in Zhu et al. (2017) we replace the log-likelihood loss with a squared difference loss which was originally introduced by Mao et al. (2016). The original motivation for this replacement was improved stability in training and avoidance of the vanishing gradients problem. In this work we found the effect on training stability substantial.

$$
\begin{aligned}
\mathcal{L}_{\text{GAN}}(F, D_{\mathcal{Y}}, \mathcal{X}, \mathcal{Y}) = {} & \mathbb{E}_{y \sim \mathcal{Y}}[(D_{\mathcal{Y}}(y \cdot W_{Emb}^{\top}))^2] \\
& + \mathbb{E}_{x \sim \mathcal{X}}[(1 - D_{\mathcal{Y}}(F(x \cdot W_{Emb}^{\top}) \cdot W_{Emb}^{\top}))^2] \\
& + \alpha \cdot \mathbb{E}_{\hat{y} \sim \hat{\mathcal{Y}}}[(1 - \|\nabla_{\hat{y}} D_{\mathcal{Y}}(\hat{y})\|_2)^2]
\end{aligned}
$$

Hence, our total loss is:

$$
\begin{aligned}
\mathcal{L}_{\text{Total}}(F, G, D_{\mathcal{X}}, D_{\mathcal{Y}}, \mathcal{X}, \mathcal{Y}) = {} & \mathcal{L}_{\text{cyc}}(F, G, \mathcal{X}, \mathcal{Y}) \\
& + \mathcal{L}_{\text{GAN}}(F, D_{\mathcal{Y}}, \mathcal{X}, \mathcal{Y}) \\
& + \mathcal{L}_{\text{GAN}}(G, D_{\mathcal{X}}, \mathcal{X}, \mathcal{Y})
\end{aligned}
$$

We adapted the convolutional architecture for the generator and discriminator directly from Zhu et al. (2017). We simply replace all two dimensional convolutions with the one dimension variant and reduce the filter sizes in our generators to 1 (pointwise convolutions). Convolutional neural networks have recently been shown to be highly effective on language tasks and can speed up training significantly (Zhang & LeCun, 2015; Kalchbrenner et al., 2016; Yu et al., 2017). Both our generators and discriminators receive a sequence of vectors in embedding space; our generators produce a softmax distribution over the vocabulary, while our discriminator produces a scalar output. For all our experiments we use a cycle loss with regularization coefficient $\lambda = 1$. In order to be compatible with the WGAN we replace batch normalization (Ioffe & Szegedy, 2015) with layer normalization (Ba et al., 2016). We train using the Adam optmizer (Kingma & Ba, 2014) with batch size 64 and learning rate $2e - 4$, $\beta_1 = 0$ and $\beta_2 = 0.9$. Our learning rate is exponentially warmed up to $2e - 4$ over 2500 steps, and held constant thereafter. We use learned embedding vectors with 256

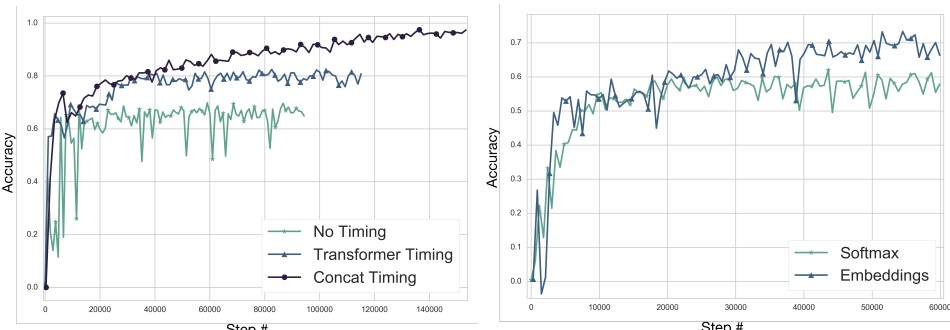

Figure 3: Left: Comparison of different timing techniques for Brown-C Vigenère. Right: Comparison of embedding vs. raw softmax on Brown-W with vocab size of 200.

| Data
Vocab size | Brown-W
10 | Brown-W
200 | Brown-C
58 | Freq. Analysis (With Key)
58 | 
200 |
|---|---|---|---|---|---|
| Cipher | **Shift/Permutation** | | | | |
| Acc. | 100% | 98.7% | 99.8% | 80.9% | 44.5% |
| Cipher | **Vigenère (Key: "345")** | | | | |
| Acc. | 99.7% | 75.7% | 99.0% | 9.6% (78.1%) | <0.1% (44.3%) |

Table 2: Average proportion of characters correctly mapped in a given sequence. The "Freq. Analysis" column is simple frequency analysis applied to the same corpus our model observes. For Vigenère we also show the score if the key were known (note: the key is left unknown to our model).

dimensions. The WGAN Lipschitz conditioning parameter was set to $\alpha = 10$ as was prescribed in Gulrajani et al. (2017).

For the Vigenère cipher, positional information is critical to the network being able to perform the mapping. In order to facilitate this we experimented with adding the timing signal described in Vaswani et al. (2017) ("Transformer Timing" in Figure 3) and found that performance increased relative to no explicit timing signal; we found that the best option was concatenating a learned positional embedding vector specific to each position onto the sequence ("Concat Timing" in Figure 3), this dramatically improved performance, however this means that the architecture can not generalize to sequences longer than those in the training set. A potential solution to the issue of generalizing to longer sequences would be making a 'soft' choice at each position for which positional embedding vector to concatenate using a softmax distribution over a set of embedding vectors larger than the expected key length, however, we leave this to future work.

## 4.3 DISCUSSION

Table 2 shows that CipherGAN was able to solve shift ciphers to near flawless accuracy, with all three vocabulary sizes being easily decoded by the model. CipherGAN performs extremely well on Vigenère, achieving excellent results on the character-level cipher and strong results on the challenging word-level cipher with a vocabulary size of 200. The vocabulary size of 58 for our character level, containing punctuation and special characters, is more than double what has been previously explored. In comparison to the original CycleGAN architecture, we found CipherGAN to be extremely consistent in training and notably insensitive to the random initialization of weights; we attribute this stability to the Jacobian norm regularization term.

For both ciphers, the first mappings to be correctly determined were those of the most frequently occurring vocabulary elements, suggesting that the network does indeed perform some form of frequency analysis to distinguish outlier frequencies in the two banks of text. Another interesting observation is that of the mistakes made by the network: the network would frequently confuse

punctuation marks with one another, perhaps suggesting that these vocabulary elements' skip-gram signatures were similar enough to lead to the repeated confusion observed across many training runs.

## 5 CONCLUSION

CipherGAN is a compelling demonstration of the potential generative adversarial networks hold to act on discrete data to solve difficult tasks that rely on an extremely sensitive and nuanced discrimination criterion. Our work serves to redouble the promise of the CycleGAN architecture for unsupervised alignment tasks for multiple classes of data. CipherGAN presents an algorithm that is both stable and consistent in training, improving upon past implementations of the CycleGAN architecture. Our work theoretically motivates – and empirically confirms – the use of continuous relaxations of discrete variables, not only to facilitate the flow of gradients through discrete nodes, but also to prevent the oft-observed phenomena of uninformative discrimination. CipherGAN is highly general in its structure and can be directly applied to a variety of unsupervised text alignment tasks, without excess burden of adaptation. On the one hand, CipherGAN is an early step towards the goal of unsupervised translation between languages and has shown excellent performance on the simplified task of cipher map inference. On the other hand, the methods we introduce can be used more broadly in the field of text generation with adversarial networks.

### ACKNOWLEDGMENTS

Our thanks goes to Roger Grosse and Kelvin Shuangjian Zhang for their advice and support throughout. We also thank Otavio Good and Ian Goodfellow for meaningful early discussions and direction; as well as Michal Wiszniewski for his assistance in developing the code upon which the experiments were run. This work was made possible thanks to the AI Grant, which provided generous support throughout.

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

APPENDIX

A   ARCHITECTURAL DETAILS

For both the generator and discriminator architectures below, we assume the network is provided with a sequence of $N$ tokens denoted `data` laying in a $K$-simplex. Therefore `data` is an $N$ by $K$ matrix.

For simplicity we define the following notation:

$$\text{ConvLayer}_{fc,fs=1,s=1}(x) = ReLU(LayerNorm(Conv1D_{fc,fs}(x)))$$

$$\text{ResBlock}_{fc,fs=1}(x) = ReLU(LayerNorm(x + Conv1D_{fc,fs}(Conv1D_{fc,fs}(x))))$$

$$\text{ConvStack}_{n,fc,fs=1}(x) =$$

$$\text{ConvLayer}_{1,fs} \circ \text{ConvLayer}_{2^1 \cdot fc,fs,2} \circ \cdots \circ \text{ConvLayer}_{2^n \cdot fc,fs,2} \circ \text{ConvLayer}_{fc,fs,2}(x)$$

A.1   GENERATOR

We define the following constants:

- $fc = 32$: The base 'filter count', or number of filters in a convolution layer
- $fs = 1$: The 'filter size', or width of weight kernel used in convolution layers
- $vs = 1$: The 'vocab size', or number of elements in the vocabulary
- $s = 1$: The stride of the convolution layers
- $E = 100$: The dimensionality of the embedding vectors
- $T = 100$: The dimensionality of the concat timing weights

First, we first look up embedding vectors corresponding to the observed tokens. This is performed as a simple inner-product between `data` and a learning weight matrix of embeddings $W_{\text{Emb}} \in \mathbb{R}^{K \times E}$:

$$x = \texttt{data} \cdot W_{\text{Emb}}$$

Next, a timing signal is added using either:

- The Transformer method:

$$\texttt{signal}_{n,2k} = \sin(n/1e5^{2k/K})$$

$$\texttt{signal}_{n,2k+1} = \cos(n/1e5^{2k/K})$$

$$\texttt{timing}(x) = x + \texttt{signal}$$

- The concat method:

$$W_{\text{time}} \in \mathbb{R}^{N \times T}$$

$$t = \texttt{timing}(x) = [x \|_2 W_{\text{time}}]$$

Where $W_{\text{time}}$ are trained parameters and $[\cdot\|_k\cdot]$ denotes concatenation along the $k^{\text{th}}$ axis.

Then the generator is defined as follows:

$$a(x) = \text{ConvLayer}_{4 \cdot fc}(\text{ConvLayer}_{2 \cdot fc,fs}(\text{ConvLayer}_{fc}(x)))$$

$$b(x) = \text{ResBlock}_{4 \cdot fc}^{(5)}(x)$$

$$\texttt{out}(t) = \text{Softmax}(\text{ConvLayer}_{vs}(b(a(t))))$$

A.2   DISCRIMINATOR

We define the following constants:

- $fc = 32$: The base 'filter count', or number of filters in a convolution layer
- $fs = 15$: The 'filter size', or width of weight kernel used in convolution layers
- $n = 5$: The depth of the convolutional stack

Similar to the generator, a timing signal is added first. Leading to the following discriminator:

$$\texttt{out}(t) = \text{ConvStack}_{n,fc,fs}(\text{dropout}_{0.5}(t))$$

## B  PROOF OF PROPOSITION 1

**Proposition 1** (Reliable Fooling Via Relaxation).
*Given:*

- *discrete spaces $\mathcal{X}, \mathcal{Y}$ and continuous relaxations $\overline{\mathcal{X}}, \overline{\mathcal{Y}}$*

- *generators $F : \overline{\mathcal{X}} \to \overline{\mathcal{Y}}, G : \overline{\mathcal{Y}} \to \overline{\mathcal{X}}$ bijections satisfying $F = G^{-1}$*

- *discrete discriminators $D_{\mathcal{X}}, D_{\mathcal{Y}}$ both optimal for fixed $F, G$*

- *rediscretization functions $R_{\mathcal{X}}, R_{\mathcal{Y}}$*

*Suppose: $F$ is approximately volume preserving in a small region about each $x \in \mathcal{X}$. Consequently the same is true for $G$ about each $y \in \mathcal{Y}$.*
*If: during training, we replace discrete random variables from $\mathcal{X}$ which lie in the continuous metric space $\overline{\mathcal{X}}$ with samples from regions about them.*
*Then: the optimal relaxed discriminators $D_{\overline{\mathcal{X}}}$ and $D_{\overline{\mathcal{Y}}}$ have a non-empty region about each $x \in \mathcal{X}$ and $y \in \mathcal{Y}$ where they are expected to assign values close to $D_{\mathcal{X}}(x)$ and $D_{\mathcal{Y}}(y)$.*

*Proof.*  We'll prove one side of the CipherGAN as the proof for both sides are similar.

Given bijective function between continuous relaxations of $\mathcal{X}$ and $\mathcal{Y}$: $F : (\overline{\mathcal{X}}, d_{\overline{\mathcal{X}}}) \to (\overline{\mathcal{Y}}, d_{\overline{\mathcal{Y}}})$, where $\mathcal{X}$ and $\mathcal{Y}$ contain finite sequences (length $n$) of vectors laying on the vertices of the simplex $\Delta^k$, and are supports of data distributions $p_{\mathcal{X}}, p_{\mathcal{Y}}$ respectively.

Let:

- $\overline{\mathcal{X}} = \overline{\mathcal{Y}} = \underbrace{\Delta^k \times \cdots \times \Delta^k}_{n}$ with $k$ equal to the number of elements in our vocabulary.

- the rediscretization function $R_{\mathcal{X}} : \overline{\mathcal{X}} \to \mathcal{X}$: $R_{\mathcal{X}}(\bar{x}) = \arg\min_{x \in \mathcal{X}} d_{\overline{\mathcal{X}}}(\bar{x}, x)$; similarly for $R_{\mathcal{Y}}$.

Now, for each $x \in \mathcal{X}$ consider the infinite set $S_x$ with cardinality of the continuum constructed according to:
$$\bar{x} \in S_x \iff \bar{x} \in \overline{\mathcal{X}}, \text{ s.t. } R_{\mathcal{X}}(\bar{x}) = x \text{ and } R_{\mathcal{Y}}(F(\bar{x})) = F(x)$$
Equivalently,
$$S_x = R_{\mathcal{X}}^{-1}(x) \cap F^{-1}(R_{\mathcal{Y}}^{-1}(F(x)))$$

**Note.** *$S_x$ is never of cardinality less than the continuum since the following is implied by the definitions of $R_{\mathcal{X}}, S_x$ and the fact that $F$ is continuous: $x \in \mathcal{X} \implies x \in S_x \wedge \exists$ closed ball $B_\epsilon[x]$ with radius*
$$0 < \epsilon < \min_{\substack{z \in \overline{\mathcal{X}} \\ R_{\mathcal{Y}}(F(z)) \neq R_{\mathcal{Y}}(F(x))}} d_{\mathcal{X}}(x, z)$$

So, for each element $x \in \mathcal{X}$ there exists a closed set of points in $\overline{\mathcal{X}}$ which are rediscretized, under $R_{\mathcal{X}}$, to $x$. Since $S_x$ is a Borel Set we can sample uniformly from it. Therefore, during training suppose we replace each element of $x \in \mathcal{X}$ with a sample $\bar{x} \sim S_x$:

We begin with the discrete objective:
$$\sum_{x \in \mathcal{X}} p_{\mathcal{X}}(x) \log(D_{\mathcal{X}}(x)) + p_F(x) \log(1 - D_{\mathcal{X}}(x))$$

As was noted in Goodfellow et al. (2014), this objective is optimized in $D_{\mathcal{X}} : \mathcal{X} \to [0, 1]$ when:
$$D_{\mathcal{X}} = \frac{p_{\mathcal{X}}}{p_{\mathcal{X}} + p_G}$$

which is undesirable as $p_{\mathcal{X}}$ is a sum of Dirac delta distributions $p_{\mathcal{X}}(x) = \sum_{x_i \in \mathcal{X}} \delta_{x_i}(x)$ and lacks a non-zero gradient to train the generator function with. Instead, let us consider a continuous relaxation $D_{\overline{\mathcal{X}}} : \overline{\mathcal{X}} \to [0, 1]$ of the discriminator $D_{\mathcal{X}}$ and observe where it optimizes.

Suppose $\forall x \in \mathcal{X}, \exists \epsilon_x \geq 0, \forall \bar{x} \in S_x$

$$(1 - \epsilon_x) \leq |\nabla_{\bar{x}} F(\bar{x})| \leq (1 + \epsilon_x) \tag{6}$$

That is, suppose $F$ is approximately volume preserving within a small region about each $x \in \mathcal{X}$.

**Lemma 1.** $\forall y \in \mathcal{Y}, G(S_y) = S_{G(y)}$

*Proof of Lemma 1.* For all $y \in \mathcal{Y}$ with $G(y) = x \Leftrightarrow y = F(x)$:

$$\begin{aligned}
G(S_y) &= G[R_{\mathcal{Y}}^{-1}(y) \cap G^{-1}(R_{\mathcal{X}}^{-1}(G(y)))] \\
&= G(R_{\mathcal{Y}}^{-1}(y)) \cap G(G^{-1}(R_{\mathcal{X}}^{-1}(G(y)))) \\
&= G(R_{\mathcal{Y}}^{-1}(y)) \cap R_{\mathcal{X}}^{-1}(G(y)) \\
&= F^{-1}(R_{\mathcal{Y}}^{-1}(F(x))) \cap R_{\mathcal{X}}^{-1}(x) \\
&= S_x = S_{G(y)}
\end{aligned}$$

$\square$

**Corollary 1.**

$$\int_{\bar{x} \sim S_{G(y)}} \frac{1}{|S_{G(y)}|} d\bar{x} = \int_{\bar{x} \sim G(S_y)} \frac{|\nabla_{\bar{x}} F(\bar{x})|}{|S_y|} d\bar{x}$$

*Proof of Corollary 1.*

$$\begin{aligned}
\int_{\bar{x} \sim S_{G(y)}} \frac{1}{|S_{G(y)}|} d\bar{x} = 1 &= \int_{\bar{y} \sim S_y} \frac{1}{|S_y|} d\bar{y} \\
&= \int_{\bar{x} \sim G(S_y)} \frac{|\nabla_{\bar{x}} G^{-1}(\bar{x})|}{|S_y|} d\bar{x} \\
&= \int_{\bar{x} \sim G(S_y)} \frac{|\nabla_{\bar{x}} F(\bar{x})|}{|S_y|} d\bar{x}
\end{aligned}$$

$\square$

**Corollary 2.**

$$\frac{1}{(1 - \epsilon_x)|S_{G(y)}|} \geq \frac{1}{|S_y|} \geq \frac{1}{(1 + \epsilon_x)|S_{G(y)}|}$$

*Proof of Corollary 2.* By Equation 6 and Corollary 1:

$$\begin{aligned}
\int_{\bar{x} \sim S_{G(y)}} \frac{1 - \epsilon_x}{|S_y|} d\bar{x} &\leq \int_{\bar{x} \sim S_{G(y)}} \frac{1}{|S_{G(y)}|} d\bar{x} \leq \int_{\bar{x} \sim S_{G(y)}} \frac{1 + \epsilon_x}{|S_y|} d\bar{x} \\
&\implies \frac{1 - \epsilon_x}{|S_y|} |S_{G(y)}| \leq 1 \leq \frac{1 + \epsilon_x}{|S_y|} |S_{G(y)}| \\
&\implies (1 - \epsilon_x)|S_{G(y)}| \leq |S_y| \leq (1 + \epsilon_x)|S_{G(y)}| \\
&\implies \frac{1}{(1 - \epsilon_x)|S_{G(y)}|} \geq \frac{1}{|S_y|} \geq \frac{1}{(1 + \epsilon_x)|S_{G(y)}|}
\end{aligned}$$

$\square$

Corollary 1 leads to the following:

$$\begin{aligned}
&\mathbb{E}_{x \in \mathcal{X}}[\mathbb{E}_{\bar{x} \in S_x}[\log(D_{\overline{\mathcal{X}}}(\bar{x}))]] + \mathbb{E}_{y \in \mathcal{Y}}[\mathbb{E}_{\bar{y} \in S_y}[\log(1 - D_{\overline{\mathcal{X}}}(G(\bar{y})))]] \\
&= \sum_{x \in \mathcal{X}} p_{\mathcal{X}}(x) \int_{\bar{x} \sim S_x} \frac{1}{|S_x|} \log(D_{\overline{\mathcal{X}}}(\bar{x})) d\bar{x} + \sum_{y \in \mathcal{Y}} p_{\mathcal{Y}}(y) \int_{\bar{y} \sim S_y} \frac{1}{|S_y|} \log(1 - D_{\overline{\mathcal{X}}}(G(\bar{y}))) d\bar{y} \\
&= \sum_{x \in \mathcal{X}} p_{\mathcal{X}}(x) \int_{\bar{x} \sim S_x} \frac{1}{|S_x|} \log(D_{\overline{\mathcal{X}}}(\bar{x})) d\bar{x} + \sum_{y \in \mathcal{Y}} p_{\mathcal{Y}}(y) \int_{\bar{x} \sim G(S_y)} \frac{|\nabla_{\bar{x}} F(\bar{x})|}{|S_y|} \log(1 - D_{\overline{\mathcal{X}}}(\bar{x})) d\bar{x}
\end{aligned} \tag{7}$$

Using Lemma 1 and Corollary 2, we obtain the following lower-bound of Equation 7:

$$\geq \sum_{x \in \mathcal{X}} p_{\mathcal{X}}(x) \int_{\bar{x} \sim S_x} \frac{1}{|S_x|} \log(D_{\overline{\mathcal{X}}}(\bar{x})) d\bar{x} + \sum_{y \in \mathcal{Y}} p_{\mathcal{Y}}(y) \int_{\bar{x} \sim G(S_y)} \frac{1 - \epsilon_x}{|S_y|} \log(1 - D_{\overline{\mathcal{X}}}(\bar{x})) d\bar{x}$$

$$\geq \sum_{x \in \mathcal{X}} p_{\mathcal{X}}(x) \int_{\bar{x} \sim S_x} \frac{1}{|S_x|} \log(D_{\overline{\mathcal{X}}}(\bar{x})) d\bar{x}$$

$$+ p_{\mathcal{Y}}(F(x)) \int_{\bar{x} \sim S_{G(F(x))}} \frac{1 - \epsilon_x}{(1 + \epsilon_x)|S_{G(F(x))}|} \log(1 - D_{\overline{\mathcal{X}}}(\bar{x})) d\bar{x}$$

$$= \sum_{x \in \mathcal{X}} \int_{\bar{x} \sim S_x} \frac{1}{|S_x|} \left[ p_{\mathcal{X}}(x) \log(D_{\overline{\mathcal{X}}}(\bar{x})) + \frac{1 - \epsilon_x}{1 + \epsilon_x} p_G(x) \log(1 - D_{\overline{\mathcal{X}}}(\bar{x})) \right] d\bar{x}$$

Which is maximal at:

$$D_{\overline{\mathcal{X}}}(\bar{x} \sim S_x) = \frac{p_{\mathcal{X}}}{p_{\mathcal{X}} + \frac{1 - \epsilon_x}{1 + \epsilon_x} p_G} \overset{\epsilon_x}{\approx} D_{\mathcal{X}}$$

$$\implies \mathbb{E}_{\bar{x} \sim S_x}[D_{\overline{\mathcal{X}}}(\bar{x})] \overset{\epsilon_x}{\approx} D_{\mathcal{X}}$$

And similarly, we find Equation 7 is upper-bounded by:

$$\leq \sum_{x \in \mathcal{X}} p_{\mathcal{X}}(x) \int_{\bar{x} \sim S_x} \frac{1}{|S_x|} \log(D_{\overline{\mathcal{X}}}(\bar{x})) d\bar{x} + \sum_{y \in \mathcal{Y}} p_{\mathcal{Y}}(y) \int_{\bar{x} \sim G(S_y)} \frac{1 + \epsilon_x}{|S_y|} \log(1 - D_{\overline{\mathcal{X}}}(\bar{x})) d\bar{x}$$

$$\leq \sum_{x \in \mathcal{X}} p_{\mathcal{X}}(x) \int_{\bar{x} \sim S_x} \frac{1}{|S_x|} \log(D_{\overline{\mathcal{X}}}(\bar{x})) d\bar{x}$$

$$+ p_{\mathcal{Y}}(F(x)) \int_{\bar{x} \sim S_{G(F(x))}} \frac{1 + \epsilon_x}{(1 - \epsilon_x)|S_{G(F(x))}|} \log(1 - D_{\overline{\mathcal{X}}}(\bar{x})) d\bar{x}$$

$$= \sum_{x \in \mathcal{X}} \int_{\bar{x} \sim S_x} \frac{1}{|S_x|} \left[ p_{\mathcal{X}}(x) \log(D_{\overline{\mathcal{X}}}(\bar{x})) + \frac{1 + \epsilon_x}{1 - \epsilon_x} p_G(x) \log(1 - D_{\overline{\mathcal{X}}}(\bar{x})) \right] d\bar{x}$$

Which is maximal in $D_{\overline{\mathcal{X}}}$ at:

$$D_{\overline{\mathcal{X}}}(\bar{x} \sim S_x) = \frac{p_{\mathcal{X}}}{p_{\mathcal{X}} + \frac{1 + \epsilon_x}{1 - \epsilon_x} p_G} \overset{\epsilon_x}{\approx} D_{\mathcal{X}}$$

$$\implies \mathbb{E}_{\bar{x} \sim S_x}[D_{\overline{\mathcal{X}}}(\bar{x})] \overset{\epsilon_x}{\approx} D_{\mathcal{X}}$$

Hence, asymptotically as $F$ becomes approximately volume-preserving about each discrete $x \in \mathcal{X}$ the bounds maximize to the same loss value in the same class of functions $\mathbb{E}_{\bar{x} \sim S_x}[D_{\overline{\mathcal{X}}}(\bar{x})] = D_{\mathcal{X}}$, application of the squeeze theorem concludes the proof. □

