# OpenReview forum: "Unsupervised Cipher Cracking Using Discrete GANs"
_ICLR.cc/2018/Conference — Accept (Poster)_

### Official Review · AnonReviewer2 · 2017-11-26
**Review for "Unsupervised Cipher Cracking Using Discrete GANs"**

**Rating:** 7
**Confidence:** 1

**Review:**

The paper shows an application of GANs to deciphering text. The goal is to arrive at a ```"hands free" approach to this problem; i.e., an approach that does not require any knowledge of the language being deciphered such as letter frequency and such. The authors start from a CycleGAN architecture, which may be used to learn mapping between two probability spaces. They point out that using GANs for discrete distributions is a challenging problem since it can lead to uninformative discriminants. They propose to  resolve this issue by using a continuous embedding space to approximate (or convert) the discrete random variables into continuous random variables. The new proposed algorithm, called CipherGAN, is then shown to be stable and achieve deciphering of substitution ciphers and Vigenere ciphers.

I did not completely understand how the embedding was performed, so perhaps the authors could elaborate on that a bit more. Apart from that, the paper is well written and well motivated. It used some recent ideas in deep learning such as Cycle GANs and shows how to tweak them to make them work for discrete problems and also make them more stable. One comment would be that the paper is decidedly an applied paper (and not much theory) since certain steps in the algorithm (such as training the discriminator loss along with the Lipschitz conditioning term) are included because it was experimentally  observed to lead to stability.

---

> ### Author Response · Authors · 2017-12-10
> **Response to R2**
>
> We thank Reviewer 2 for their helpful comments.
>
> Thank you for pointing out the lack of clarity w.r.t. the embeddings. We have added more discussion about precisely how we treat the embeddings.
>
> We hope that the reviewer will be able to assign more confidence to their review given the changes. Please inform us of any further changes that would improve the paper.
>
> Again, we sincerely thank the reviewer for their time and support.

---

### Official Review · AnonReviewer3 · 2017-11-27
**This paper proposed CipherGAN method addressing shift and Viegenere ciphers. The performance is better than CycleGAN and relatively stable under random initial weights. This well written paper adds value to decoding literature.**

**Rating:** 8
**Confidence:** 4

**Review:**

The paper proposed to replace the 2-dim convolutions in CycleGAN by one dimension variant and reduce the filter sizes to 1, while leave the generator convex embedding and using L2 loss function.

The proposed simple change help with the dealing of discrete GAN. The benefit of increased stability by adding Jacobian norm regularization term to the discriminator's loss is nice.

The paper is well written. A few minor ones to improve:
* The original GAN was proposed/stated as min_max, while in Equation 1 didn't defined F and was not clear about min_{F}. Similar for Equations 2 and 3.
* Define abbreviation when first appear, e.g. WGAN (Wasserstein ...).
* Clarify x- and y- axis label in Figure 3.

---

> ### Author Response · Authors · 2017-12-10
> **Response to R3**
>
> We thank Reviewer 3 for their suggested improvements.
>
> We have incorporated all three suggestions into the latest draft of the paper. Please inform us of any changes that would further improve the work.
>
> Thank you again for your review.

---

### Official Review · AnonReviewer1 · 2017-11-30
**Interesting but the mismatch between theory and experiments is an issue and some points of the proof need to be clarified...**

**Rating:** 7
**Confidence:** 4

**Review:**

SUMMARY

The paper considers the problem of using cycle GANs to decipher text encrypted with historical ciphers. Also it presents some theory to address the problem that discriminating between the discrete data and continuous prediction is too simple. The model proposed is a variant of the cycle GAN in which in addition embeddings helping the Generator are learned for all the values of the discrete variables.
The log loss of the GAN is replaced by a quadratic loss and a regularization of the Jacobian of the discriminator. Experiments show that the method is very effective.

REVIEW

The paper considers an interesting and fairly original problem and the overall discussion of ciphers is quite nice. Unfortunately, my understanding is that the theory proposed in section 2 does not correspond to the scheme used in the experiments (contrarily to what the conclusion suggest and contrarily to what the discussion of the end of section 3, which says that using embedding is assumed to have an equivalent effect to using the methodology considered in the theoretical part). Another important concern is with the proof: there seems to be an unmotivated additional assumption that appears in the middle of the proof of Proposition 1 + some steps need to be clarified (see comment 16 below).
The experiments do not have any simple baseline, which is somewhat unfortunate.


DETAILED COMMENTS:

1- The paper makes a few bold and debatable statements:

line 9 of section 1
"Such hand-crafted features have fallen out of favor (Goodfellow et al., 2016) as a
result of their demonstrated inferiority to features learned directly from data in end-to-end learning
frameworks such as neural networks"

This is certainly an overstatement and although it might be true for specific types of inputs it is not universally true, most deep architectures rely on a human-in-the-loop and there are number of areas where human crafted feature are arguably still relevant, if only to specify what is the input of a deep network: there are many domains where the notion of raw data does not make sense, and, when it does, it is usually associated with a sensing device that has been designed by a human and which implicitly imposes what the data is based on human expertise.

2- In the last paragraph of the introduction, the paper says that previous work has only worked on vocabularies of 26 characters while the current paper tackles word level ciphers with 200 words. But, isn't this just a matter of scalability and only possible with very large amounts of text? Is it really because of an intrinsic limitation or lack of scalability of previous approaches or just because the authors of the corresponding papers did not care to present larger scale experiments?


3- The discussion at the top of page 5 is difficult to follow. What do you mean when you say "this motivates the benefits of having strong curvature globally, as opposed to linearly between etc"
Which curvature are we talking about? and what how does the "as opposed to linearly" mean? Should we understand "as opposed to having curvature linearly interpolated between etc" or "as opposed to having a linear function"? Please clarify.

4- In the same paragraph: what does "a region that has not seen the Jacobian norm applied to it" mean? How is a norm applied to a region? I guess that what you mean is that the generator G might creates samples in a part of the space where the function F has not yet been learned and is essentially close to 0. Is this what you mean?

5- I do not understand why the paper introduces WGAN since in the end it does not use them but uses a quadratic loss, introduced in the first display of section 4.3.

6- The paper makes a theoretical contribution which supports replacing the sample y by a sample drawn from a region around y. But it seems that this is not used in the experiment and that the authors consider that the introduction of the embedding is a substitution for this. Indeed, in the last paragraph of section 3.1, the paper says "we make the assumption that the training of the embedding vectors approximates random sampling similar to what is described in Proposition 1". This does not make any sense to me because the embedding vectors map each y deterministically to a single point, and so the distribution on the corresponding vectors is still a fixed discrete distribution. This gives me this impression that the proposed theory does not match what is used in the experiments.
(The last sentence of section 3.1, which is commenting on this and could perhaps clarify the situation is ill formed with two verbs.)

7- In the definitions: "A discriminator is said to perform uninformative discrimination" etc. -> It seems that the choice of the word uninformative would be misleading: an uninformative discrimination would be a discrimination that completely fails, while what the condition is saying it that it cannot perform perfect discrimination. I would thus suggest to call this "imperfect discrimination".


8- It seems that the same embedding is used in X space and in Y space (from equations 6 and 7). Is there any reason for that? I would seem more natural to me to introduce two different embeddings since the objects are a priori different...
Actually I don't understand how the embeddings can be the same in the Vignere code case since time taken into account one one side.

9- On the 5th line after equation (7), the paper says "the embeddings... are trained to minimize L_GAN and L_cyc, meaning... and are easy to discriminate" -> This last part of the sentence seems wrong to me. The discriminator is trying to maximize L_GAN and so minimizing w.r.t. to the embedding is precisely trying to prevent to the discriminator to tell apart too easily the true elements from the estimated ones.
In fact the regularization of the Jacobian that will be preventing the discriminator to vary too quickly in space is more likely to explain the fact that the discrimination is not too easy to do between the true and mapped embeddings. This might be connected to the discussion at the top of page 5. Since there are no experiments with alpha different than the default value = 10, this is difficult to assess.

10-The Vigenere cipher is explained again at the end of section 4.2 when it has already been presented in section 1.1

11- Concerning results in Table 2: I do not see why it would not be possible to compare the performance of the method with classical frequency analysis, at least for the character case.

12- At the beginning of section 4.3, the text says that the log loss was replaced with the quadratic loss, but without giving any reason. Could you explain why.

13- The only comparison of results with and without embeddings is presented in the curves of figure 3, for Brown-W with a vocabulary of 200 words. In that case it helps. Could the authors report systematically results about all cases? (I guess this might however be the only hard case...)

14- It would be useful to have a brief reminder of the architecture of the neural network (right now the reader is just refered to Zhu et al., 2017): how many layers, how many convolution layers etc.
The same comment applies for the way the position of the letter/word in the text appear is in encoded in a feature that is provided as input to the neural network: it would be nice if the paper could provide a few details here and be more self contained. (The fact that the engineering of the time feature can "dramatically" improve the performance of the network should be an argument to convince the authors that hand-crafted feature have not fallen out of favor completely yet...)

15- I disagree with the statement made in the conclusion that the proposed work "empirically confirms [...] that the use of continuous relaxation of discrete variable facilitates [...] and prevents [...]" because for me the proposed implementation does not use at all the theoretical idea of continuous relaxation proposed in the paper, unless there is a major point that I am missing.


16- I have two issues with the proof in the appendix

a) after the first display of the last page the paper makes an additional assumption which is not announced in the statement of the theorem, which is that two specific inequality hold...
Unless I am mistaken this assumption is never proven (later or earlier). Given that this inequality is just "the right inequality to get the proof go through" and given that there are no explanation for why this assumption is reasonable, to me this invalidates the proof. The step of going from G(S_y) to S_(G(y)) seems delicate...

b) If we accept these inequalities, the determinant of the Jacobian (the notation is not defined) of F at (x_bar) disappears from the equations, as if it could be assumed to be greater than one. If this is indeed the case, please provide a justification of this step.

17- A way to address the issue of trivial discrimination in GANs with discrete data has been proposed in

Luc, P., Couprie, C., Chintala, S., & Verbeek, J. (2016). Semantic segmentation using adversarial networks. arXiv preprint arXiv:1611.08408.
The authors should probably reference this paper.


18- Clarification of the Jacobian regularization: in equation (3), the Jacobian computed seems to be w.r.t D composed with F while in equation (8) it is only the Jacobian of D. Which equation is the correct one?

TYPOS:

Proposition 1: the if-then statement is broken into two sentences separated by a full point and a carriage return.

sec. 4.3 line 10 we use a cycle loss *with a regularization coefficient* lambda=1 (a piece of the sentence is missing)

sec. 4.3 lines 12-13 the learning rates given are the same at startup and after "warming up"...

In the appendix:
3rd line of proof of prop 1: I don' understand "countably infinite finite sequences of vectors lying in the vertices of the simplex" -> what is countable infinite here? The vertices?

---

> ### Author Response · Authors · 2017-12-10
> **Response to R1**
>
> We would like to thank Reviewer 1 for providing such a high quality and clear review that has allowed us to greatly improve our paper. We hope the new draft of the paper and the clarifications and improvements made below serve to increase your rating of our work.
>
> We address each point below:
>
> 1. We completely agree that this was an overstatement and have replaced the line with the following "Across a number of domains, the use of hand-crafted features has often been replaced by automatic feature extraction directly from data using end-to-end learning frameworks (Goodfellow et al., 2016)." We qualify the statement, restricting it to ‘a number of’ domains of application, while acknowledging that automated feature extraction is not ubiquitous and removing any notion of superiority/inferiority of techniques.
>
> 2. The issue of scalability is indeed an important one; as past algorithms have scaled exponentially in the length of key and size of vocabulary. Prior work has generally relied upon ngram frequencies whose space grows exponentially with the vocabulary size rapidly sparsifying occurrences; leading to rapidly decreasing information in these statistics. The other facet of increasing vocabulary size is the applicability to more modern techniques such as block ciphers where the vocabulary is expanded to hundreds or thousands of elements. We intended to show that our method doesn’t completely collapse as vocab space increases (while frequency analysis rapidly does, as we show in new baseline comparisons). We feel this is a valuable and important feature of the work.
>
> 3. We clarified the discussion making things a little less verbose and trying to improve the flow of ideas. The curvature we refer to is that of the discriminator output with respect to its inputs (we’ve tried to clarify this); the curvature of this region is important since it represents the strength of the training signal received by the generator. We were trying to make the point that WGAN’s method of regularizing between the generated data and the true data may miss regions of the simplex that our model regularly traverses. We also note that others have pointed this out and have found benefits of regularizing more ‘globally’ or more ‘broadly’ across the simplex (by this we mean other than exclusively between generated data and true data; we have tried to make this clearer in the paper). We hope the changes are an improvement and that it reads more intelligibly. Thank you for raising this concern.
>
> 4. Thank you for catching this, that was indeed a typo. We’ve corrected to clarify that it is the curvature regularization being applied (i.e. the WGAN regularization technique of forcing the norm of the Jacobian to 1)
>
> 5. We introduce WGAN because it is the inspiration for the last term of our L_GAN loss. The key insight we draw from WGAN is their use of a Jacobian norm regularization term (we also refer to it as ‘curvature regularization’ since it is clearer).
>
> 6. Thank you for pointing this out; we’ve done our best to make the precise use in our paper clear. The embeddings are not fixed during training, instead, they are parameters that are tuned throughout training. It is the stochasticity of these points that leads us to the suggestion that these points estimate random samples around fixed points. We came to this conclusion after observing that, as training progresses, the embeddings appear to ‘settle’ and remain bound within a tight region, yet are still moving. Perhaps an analogy to Hamiltonian MCMC or Metropolis-adjusted Langevin sampling as a comparison between noisy gradient updates and gradient-based sampling would improve the argument? We’ve updated the paper to include a clearer motivation of why we suggest jointly-trained embedding vectors might approximate sampling about fixed points. We’ve updated the last sentence of section 3.1 to clarify precisely how we arrived at our conclusion.
>
> 7. We refer to this behaviour of the discriminator as uninformative since it says: if we can re-discretize an element to the correct token, but the discriminator evaluates it as incorrect, then the discriminator is not informing on the underlying task when acting on the continuous space. We don’t mean to say that it is ‘entirely’ uninformative of task, only that it demonstrates uninformative behaviour. We are willing to update the name to ‘partially uninformative discrimination’ if the reviewer feels this is an abuse of language?
>
> Continued in next comment.

---

> > ### Author Response · Authors · 2017-12-10
> > **Continued Response to R1**
> >
> > 8. Indeed, we do use the same space for X and Y (it’s only the distribution over the spaces that changes). This is possible since in shift and Vigenere we replace each token of the input space with a different token from the same space. I.e. “ABC” -> “GHI” for a shift or “ABC” -> “GGI” for Vigenere with key “656”. This is why we use the same embeddings for both the plaintext data space and ciphertext space. Perhaps we should note that we did experiment with separating the embedding spaces and found little improvement?
> >
> > 9. We have corrected to ‘maximizing’ L_GAN. We analyze the effects of the Jacobian regularization effects in Section 2 as well as in Figure 3 (right). You are correct that Jacobian regularization certainly helps with the problem and we cite three papers which mention this; but in our experiments (see Figure 3 comparison between embeddings and softmax) we found that Jacobian regularization was complemented by our relaxed sampling technique.
> >
> > 10. Thank you for pointing this out, we have removed Section 4.2
> >
> > 11. We completely agree that comparison to standard frequency analysis should be shown and have added this to the table. As is made clear, CipherGAN out-performs frequency analysis by a large margin (>20%). CipherGAN was able to crack all ciphers to nearly flawless accuracy (save Vigenere Brown 200, which is an extremely difficult case we use to stress test the technique).
> >
> > 12. Absolutely; we have added some of the details from Mao et al.
> >
> > 13. You are correct, the only relevant experiment was on Vigenere with Brown 200 since it challenged the network’s ability the most and exposed the divergence in performance between the two techniques.
> >
> > 14. We have added a full description of the architecture in the Appendix.
> >
> > 15. Hopefully the previous clarification resolves this critique.
> >
> > 16. Both your points are correct, the previous version of the paper had a proof that was ‘in between’ two directions. One being an analogue to Ian’s proof in the original GAN paper, and the other being an asymptotic argument that ended up being more elegant and easy to follow. In the updated paper we hope you find the new proof clearly articulated and thoroughly justified. Your point in a) about G(S_y) to S_(G(y)) appears in Lemma 1;  the note about inequalities is clarified using Corollary 2; the note about the Jacobian is now stated in the premises of the proposition and we now only require the Jacobian to be near 1 and show that as it approaches 1 the upper and lower bounds squeeze to the same maximal value in the same place.
> >
> > 17. Thank you, we have added a citation in the discrete gan section.
> >
> > 18. Very good catch, thank you. We have correct Eq. 3
> >
> > We have also addressed the typos pointed out by the reviewer.
> >
> > The reviewer’s principal concern seems to stem from the assumption of embeddings approximating sampling. We hope our clarification that our embeddings are non-fixed points and that experiments with Concrete samples produce nearly indistinguishable results give the review confidence in our methods. Additionally we hope the new proof convinces the reviewer and address the previous concerns (which arose from the proof being incomplete at the time of submission). We hope that the reviewer finds confidence in both the theoretical contributions and the success of the experiments in order to raise the rating to one of acceptance.
> >
> > Again, we sincerely appreciate such a detailed and exemplary critique of our work. Please inform us of any other changes that would improve our work.

---

> > > ### Comment · AnonReviewer1 · 2018-01-12
> > > **Response to rebuttal**
> > >
> > > I appreciate the responses to my points made in the rebuttal. They address my various concerns quite well and the updated version of the paper is quite compelling.
> > >
> > > Here are a few comments in response to the rebuttal:
> > >
> > > I found the response to my point 2 quite interesting and worth including in the paper.
> > >
> > > About point 6: Thank you for this explanation. If I understand correctly, you are trusting that the stochastic updates "simulates" some random sampling of the points. But in that case why not considering also the method corresponding exactly to the theory in which the embeddings would actually be sampled just once in a neighborhood as preprocessing, as opposed to being learned (or yet resampled at each visit of a datapoint). I would seem a reasonable baseline + a reasonable validation of the theory...
> > >
> > > 7. In my opinion: "Partially uninformative" sounds good. "Uninformative" sounds a bit too strong. But I would not want to necessarily impose this.
> > >
> > > 8. Yes, I would find it useful to have the information that you also tested the formulation with different embeddings spaces.
> > >
> > > In terms of the proof in appendix B, I would find useful to have a discussion of why Assumption of equation (6) is reasonable...

---

### Decision · Program_Chairs · 2018-01-29
**ICLR 2018 Conference Acceptance Decision**

**Decision:**

Accept (Poster)

**Comment:**

this work adapts cycle GAN to the problem of decipherment with some success. it's still an early result, but all the reviewers have found it to be interesting and worthwhile for publication.